

# Multi-static spatial and angular studies of polar mesospheric summer echoes combining MAARSY and KAIRA

Jorge. L. Chau[1], Derek McKay[2,3], Juha P. Vierinen[2], Cesar La Hoz[2], Thomas Ulich[3], Markku Lehtinen[3], and Ralph Latteck[1]

[1]Leibniz Institute of Atmospheric Physics at the University of Rostock, Kühlungsborn, Germany
[2]UiT, The Arctic University of Norway, Tromsø, Norway
[3]Sodankylä Geophysical Observatory, Sodankylä, Finland

*Correspondence to:* Jorge L. Chau (chau@iap-kborn.de)

**Abstract.** Polar mesospheric summer echoes (PMSEs) have been long associated with Noctilucent clouds (NLCs). For large ice particles sizes and relatively high ice densities, PMSE and NLCs have been shown to be highly correlated at 3-m Bragg wavelengths and are known to be good tracers of the atmospheric wind dynamics. Combining the Middle Atmosphere ALO-MAR Radar System (MAARSY) and the Kilpisjärvi Atmospheric Imaging Receiver Array (KAIRA), i.e., monostatic and bistatic observations, we show for the first time direct evidence of limited-volume PMSE structures drifting more than 90 km almost unchanged. These structures are shown to have widths of 5-15 km and are separated by 20-60 km, consistent with structures due to atmospheric waves previously observed in NLCs from the ground and from space. Given the lower sensitivity of KAIRA, the observed features are attributed to echoes from regions with high Schmidt numbers that provide a large radar cross-section. The bistatic geometry allows us to determine an upper value for the angular sensitivity of PMSE echoes at meter scales. We find no evidence for strong aspect sensitivity for PMSE echoes, which is consistent with recent observations using radar imaging approaches. Our results indicate that multi-static all-sky interferometric radar observations of PMSE could be a powerful tool for studying mesospheric wind-fields within large geographic areas.

*Copyright statement.* TEXT

## 1 Introduction

The strong radar echoes over latitudes during the summer were first reported by Ecklund and Balsley (1981). Since then these echoes, known as Polar Mesospheric Summer Echoes (PMSE) (e.g., Hoppe et al., 1988), have been the subject of active research. Currently there is a general consensus that they are generated by atmospheric turbulence and require the presence of free electrons and charged ice particles (e.g., Kelley and Ulwick, 1988; Havnes et al., 1996; Lie-Svendsen et al., 2003b, a; Rapp and Lübken, 2004). The scattering theories have been improved in the last decade to include events that were not supported before. For example, Varney et al. (2011) improved the previous work of Rapp et al. (2008) to explain the PMSE observations with the Poker Flat Incoherent Scatter Radar (PFISR). Namely, they arrived to an expression that shows that PMSE radar





cross section (RCS) depends on electron density when it is much smaller than ice density. In the reverse case, PMSE RCS is proportional mainly to ice density.

The connection between noctilucent clouds (NLCs) and PMSE has been established by many authors (e.g. Hoppe et al., 1990; Nussbaumer et al., 1996; Stebel et al., 2000; Kaifler et al., 2011). The common element in PMSE and NLCs is the

presence of ice particles in the summer polar mesosphere. A difference is that only the electrically charged population of the particles have a role in the radar scattering mechanism regardless of their size. On the other hand, only particles of size greater than about 40 nm contribute to the brightness of NLCs regardless of whether they are charged or not. Another important difference is that NLCs cover a large part of the sky and they are visible even to the naked eye, which facilitates the study of their large-scale horizontal behaviour with various types of all-sky cameras. In contrast, PMSE can be studied only inside a very

limited region determined by the radar antenna beamwidth, typically a few km wide in the transverse (horizontal) direction. Thus, NLCs provide means to investigate the large scale behaviour of the polar mesosphere. Observations with a variety of optical instruments have shown that NLCs present a variety of horizontal scales, from meters to hundreds of kilometers and are typically confined to layers of less than 1 km thickness (e.g. Fiedler et al., 2009; Baumgarten and Fritts, 2014). From the temporal and spatial evolution of these structures, atmospheric waves and instabilities can be studied (e.g., Fritts et al., 2014,

and references therein).

Given the understanding of PMSE occurrence, recent efforts have been devoted to study their long term behavior (Latteck and Bremer, 2017, e.g.,), their angular dependence (e.g., Czechowsky et al., 1988; Huaman and Balsley, 1998; Smirnova et al., 2012; Sommer et al., 2016), and using them as tracers for atmospheric dynamics (e.g. Balsley and Riddle, 1984; Fritts et al., 1990; Hoppe and Fritts, 1995; Stober et al., 2013). Following the relation with NLCs, simultaneous PMSE observations with

spatially separated monostatic systems and special atmospheric conditions have been associated to drifting structures (e.g., Bremer et al., 1996; Belova et al., 2007; Rapp et al., 2008). However, these previous studies were not able to determine the size and separation of such drifting structures. Recently, Sommer and Chau (2016) using radar imaging have reported PMSE horizontal structures with sizes around 1 km, which is not surprising given that even smaller structures have been observed in NLCs. Moreover, these findings support the hypothesis of Sommer et al. (2016) that the radar cross-section of PMSE does not

vary significantly as a function of observing angle (aspect sensitivity). This implies that the scattering originates from localized isotropic structures, instead of anisotropic horizontally stratified structures. If the high aspect sensitivity were the norm for PMSE, our observations reported here would not have been possible.

In this paper we present the results obtained using the Middle Atmosphere ALOMAR Radar System (MAARSY) (16.04°E, 69.30°N) and the Kilpisjärvi Atmospheric Imaging Receiver Array (KAIRA) (20.76°E, 69.07°N) in northern Scandinavia.

MAARSY is a powerful all-digital phase array radar that was specially built to study PMSE and lower atmospheric dynamics (Latteck et al., 2012b). KAIRA was designed to study different kinds of atmospheric and ionospheric phenomena (Vierinen et al., 2013; McKay-Bukowski et al., 2015). KAIRA can be used as an all-sky imaging receiver for cosmic radio emissions to study D-region absorption (McKay et al., 2015), and as a phased array radar receiver (Virtanen et al., 2014) for nearby radar and radio transmitters, such as the EISCAT VHF radar (European Incoherent Scatter Scientific Association), MAARSY, or

several nearby specular meteor radars.





Our paper is organized as follows. We first cover some aspects of PMSE scattering theory with special emphasis on meter scales and Bragg wavelength dependence. Then we describe the experiment configuration and present important aspects of the bistatic geometry, e.g., the effective Bragg wavelength. The monostatic and bistatic experimental results are shown and combined in Section 4. We proceed to discuss the horizontal sizes and separations of the identified structures and the conservative

angular dependence values derived. Finally we present our conclusions emphasizing the possibility of using observations of this kind to both gain more insight on PMSE spatial-temporal features and to potentially use PMSE scattering as a way of obtaining improved regional wind field measurements.

## 2   PMSE scattering theory at meter scales

The expected radar cross-section of PMSE has been studied by many authors trying to explain observations at different radar

frequencies under different natural as well as artificial (ionospheric modification using HF radio waves) conditions (e.g., Hill et al., 1999; Rapp and Lübken, 2004; La Hoz et al., 2006; Rapp et al., 2008; Varney et al., 2011). The most recent work in the subject by Varney et al. (2011), following the work of Hill (1978) and Rapp et al. (2008), shows that RCS is a strong function of electron density only when electron density is much smaller than ice density. Otherwise it is mainly controlled by ice density. This improvement to previous theories was motivated by PMSE observations with the Poker Flat Incoherent Scatter Radar

(PFISR) during night and during aurora.

To help in the presentation and interpretation of our results, here we briefly show expressions relevant for our Bragg wavelengths of interest, i.e., around 3 meters. From EQ 44 in Varney et al. (2011), the PMSE RCS as function of Bragg wavenumber, i.e., $k_B = 2\pi/\lambda_B$ , is

$$\eta(\boldsymbol{k}_B) \propto k_B^{-3} \exp\left(\frac{-qK_\kappa k_B^2}{S_c}\right) \tag{1}$$

where $Sc = \nu/D$ is the Schmidt number, $K_\kappa = (\nu_a^3/\epsilon)^{1/4}$ is the Kolmogorov microscale, $\nu_a$ is the kinematic viscosity of air, $D$ the diffusion coefficient of electrons, and $\epsilon$ is the energy dissipation rate of turbulence. For a turbulent velocity spectrum with a Gaussian shape and width (or turbulence intensity) $\sigma_v$, $\epsilon = F\sigma_v^2$, where $F$ is factor that varies typically between 8 and 10 depending on the actual atmospheric conditions (e.g., Hocking, 1985; La Hoz et al., 2006).

The expected dependence of PMSE RCS at meter scales is shown in Figure 1a. The figure shows the expected PMSE RCS

for two simulations as a function of $k_B$. The simulations have been obtained with a model used by La Hoz et al. (2006), which is based on the seminal work of Hill (1978). Moreover, we have used model 2 of Hill (1978) as suggested by Hill et al. (1999). In the first simulation, we keep $S_c = 3$ constant and vary the turbulence intensity ($\sigma_v$) (dashed lines). In the second simulation, we fix $\sigma_v = 1.1$ m/s and vary $Sc$. The vertical dashed-dot-dashed lines represent the lower and larger (MAARSY monostatic) wavenumbers we will explore in this study with bistatic and monostatic radar experiments. The black solid and dashed curves

represent Bragg wavenumber dependence shown in EQ 1. Note that for high Schmidt number RCS is independent of turbulence intensity, but has a clear dependence on $\lambda_B^3$ for the Bragg wavelengths of interest.





In all cases, the simulations have been conducted for typical PMSE altitudes (i.e., 85 km). We have assumed $N_e = 3. \times 10^{-9}$, $h_H = 0.2$, $\nu = 0.567$, $D_{ne} = 0.567$, $\sigma_{ne} = 1000$, and $\chi = 2.6 \times 10^{12}$. Here $N_e$ is electron density in e m$^{-3}$, $\nu$ atmospheric viscosity in m$^2$s$^{-1}$, $D_e$ electron diffusion rate in m$^2$s$^{-1}$, $\sigma_{ne}$ scale length of an electron density byte-out in m, and $\chi$ the dissipation rate of electron density variance in m$^{-6}$s$^{-1}$. $h_H$ is the Havnes parameter given by $Z_d N_d / N_e$ (Verheest, 2000),

where $Z_d$ is the charge dust density and $N_d$ the dust number density.

Figure 1b shows relative RCSs as a function of turbulence intensity for different simulations at MAARSY's wavenumber in a monostatic configuration, i.e., $2\pi/2.8$, namely for: (a) a large $S_c 0$ (900) (green), (b) a small $S_c$ (3) (dashed black), (c) turbulence without ice at 70 km (orange), (d) a fix $\sigma_v = 1.1$ for a wide range of $S_c$ (blue), and (e) a fix $\sigma = 4.0$ for large $S_c$ (from 100 to 5000) (red triangles). We have intentionally removed the absolute RCS from Figure 1b, since we want to

emphasize the qualitative features of these results.

The salient features at about 2.8 m Bragg wavelength that can be deduced from Figure 1b are:

1. Once the $S_c$ is high (e.g., $> 100$), the RCS varies very little (red triangles).

2. The RCS varies significantly as function of $S_c$ when $S_c$ is not too large (e.g., $Sc < 100$). In our simulation with $\sigma_v = 1.1$ (blue) the variation is more than 6 orders of magnitude.

3. At low $S_c$, the RCS varies strongly with turbulence intensity (dashed black), in a manner similar to turbulence without ice. As a reference, we are showing the expected RCS with $S_c = 1$ but at 70 km instead of 85 km (orange). Note the increase of RCS with increasing $\sigma_v$.

4. At high $S_c$, the RCS decreases with turbulence intensity (green line). This is an unexpected result, since this behavior can not be reproduced by expressions or results shown in previous works (e.g., Rapp et al., 2008; Varney et al., 2011).

This result indicates that for a given Schmidt number there is a region, $k_B \gg k_T$, where the RCS does increase with increasing turbulence intensity, and for $k_B \ll k_T$ where the RCS decreases with increasing turbulence intensity, where $k_T$ is the Bragg wavenumber of this transition. Note that our simulations have been obtained by numerically integrating Hill's theoretical results without approximations.

## 3   Experiment Description

As mentioned above, the results presented in this paper have been obtained using MAARSY and KAIRA in northern Scandinavia. The distance between the two systems is approximately 190 km. MAARSY was used for transmission and reception operating at 53.5 MHz, i.e., a radar wavelength of 5.61 m. KAIRA was used for reception only. Figure 2 shows an schematic view of the experiment. As reference, we show the directions of the Bragg vectors of the bistatic geometry, i.e., KAIRA receptions, with white arrows, which all point to the middle point between MAARSY and KAIRA. Note that the Bragg wavelengths

will be different for different vectors, being the largest over the middle point (i.e., $\sim 4.15$ m) and the smallest for a monostatic configuration (i.e., 2.8 m). Below we describe the specific configuration for each system as well as the main geometrical parameters of the MAARSY-KAIRA configuration. In a conventional bistatic configuration without scanning (as is thae case



here) only one Bragg vector contributes to the received bistatic signal. The KAIRA data gathered during this experiment shows otherwise, as the received signals at KAIRA have contributions originating from the MAARSY's antenna sidelobes in a wide range of directions, each with different Bragg wavevectors represented by the white arrows in Figure 2; see below.

### 3.1 MAARSY Configuration

MAARSY consists of 433 crossed-polarized 3-element Yagi antennas. On transmission right-circular polarization is used; the beam can be steered from pulse-to-pulse every 1 ms; and different sections of the antenna can be used. On reception there are 16 complex channels available. One of these channels receives from all 433 antenna elements, while the other 15 can be selected to receive from different portions of the antenna. General details of the system are given by Latteck et al. (2012a). An example of MAARSY's flexibility on transmission and reception can be found in Sommer and Chau (2016), where narrow and wide beams were used on transmission, and 15 different groups of 7 antennas each (called hexagons) were used on reception.

For this campaign MAARSY was run with a complementary code using 2 $\mu$s baudwidth, 5% duty cycle of the available power, and an interpulse period of 1 ms. Only one vertically-pointing direction was used on both transmission and reception. Complex voltages for the added signal of all 433 elements were recorded. To allow synchronization with KAIRA, a 1 pulse-per-second GPS pulse and a GPS-disciplined rubidium clock were used. The data was analyzed offline to obtain spectra and spectral moments.

### 3.2 KAIRA Configuration

KAIRA is a dual array of omnidirectional VHF radio antennas in northern Finland. It consists of two closely located arrays working in the bands between 10 and 80 MHz and between 110 and 250 MHz, using LOFAR antenna and digital signal-processing hardware. Here we have used the former which is called the Lower Band Array (LBA). The LBA consists of 48 crossed inverted-V-dipole antennas. Each of the signal channels, i.e., 96 including the two linear polarizations, are directly sampled. After sampling the signals are processed and combined in a variety of possibilities that could combine frequency bands, antenna elements, and antenna pointing directions, each such configuration referred to as a "beamlet". The specific characteristics of KAIRA as well as results as stand-alone and as receiver for other transmitters can be found in McKay-Bukowski et al. (2015).

The part of the experiment that we used in this paper consisted of 5 beamlets all of them using all 48 LBA antennas pointing over MAARSY, i.e.,-80° azimuth, and 68.20° zenith, with different center frequencies around 53.55 MHz, and a frequency width of 195.312 kHz, i.e., allowing an effective sampling of $\sim$1 $\mu$s. Complex voltages for each beamlet were recorded and later combined, decoded, arranged in range, and spectrally analyzed.

The whole experiment during this campaign consisted of 61 beamlets: 14 beamlets using 7 single selected antennas and 2 subbands around 32.55 MHz, 35 beamlets using the same 7 antennas as before but with 5 subbands around 53.5 MHz, 10 beamlets using two pointing directions over MAARSY and 5 subbands around 53.5 MHz, and 2 beamlets in riometer mode using two different pointing directions. The experiment was conducted for almost three days around August 12, 2016. The main purpose of the experiment was to apply the MMARIA (Multi-static, Multi-frequency Agile Radar Investigations of the



Atmosphere) (e.g., Stober and Chau, 2015; Chau et al., 2017) in KAIRA, using MAARSY and the Andenes specular meteor radar working at 32.55 MHz as transmitters, respectively. Unfortunately, for the purpose of this work, only 5 and half hours of the subbands around MAARSY were recorded, mainly due to the high data volume. 14 terabytes of data were recorded during these three days. The results related to the MMARIA approach and the 32.55 MHz will be left for a future effort.

## 3.3   Bistatic Geometry and Considerations

The scattering of interest will be given by the Bragg wavelength components, i.e., $\lambda_B$, where $|\mathbf{k}_B| = 2\pi/\lambda_B$ and $\mathbf{k}_B = \mathbf{k}_s - \mathbf{k}_i$ and $\mathbf{k}_i$ and $\mathbf{k}_s$ are the incident and scattered wavenumbers with magnitudes $2\pi/\lambda$, where $\lambda$ is the radar wavelength. The Bragg wavelength and the radar wavelength are related by $\lambda_B = \lambda/(2\cos(\theta_B/2))$, where $\theta_B$ is the scattering angle, i.e., the angle between $\mathbf{k}_i$ and $\mathbf{k}_s$.

In Figure 3, we show contour plots of selected parameters of the bistatic geometry at an altitude of 85 km, as function of longitude and latitude. Specifically, we show: (a) the normalized antenna gain of MAARSY, (b) the normalized antenna gain of KAIRA, (c) the total range, (d) the Bragg wavelength, (e) a horizontal distance, and (f) the local scattering angle, i.e., the angle with respect to the local coordinate system taking into account the geoid form of the Earth. By total range we mean the distance from transmitter to scattering center plus the distance from scattering center to receiver. In monostatic systems the range to
the scattering center is half the total range. The MAARSY and the middle point between MAARSY and KAIRA are indicated by a square and a triangle respectively. Note that, contrary to monostatic configurations, the contours are not symmetric with respect to the center point.

A simple version of the radar equation assuming that the target fills the radar beam, satisfies the Born approximation, and its located in the far-field, is given by:

$$P_r = P_t \frac{G_t}{4\pi R_i^2} V\eta \frac{A_r}{4\pi R_s^2} \sin^2 \delta \qquad (2)$$

where $\eta$ is the radar scattering cross section, $P_r$ is the received power, $P_t$ is the transmitted power, $G_t$ the transmitted antenna gain, $A_r$ the receiver effective antenna area, $V$ is the scattering volume, $\delta$ is the polarization angle, and $R_i, R_s$ are the incident and scattered ranges, respectively. Given that on transmission a right-circular polarization was used, and on reception the power of two orthogonal linear polarizations were employed, $\sin^2 \delta = 0.5 + 0.5\cos^2 \theta_B$.

Taking into account the antenna patterns and replacing $A_r = G_r\lambda^2/(4\pi)$, the bistatic backscatter power at a given total range $R_0$ is given by

$$P_r(R_0) = P_t \frac{\lambda^2}{16\pi^2} \frac{1}{8\pi} \int\limits_{R_0-c\tau/2}^{R_0+c\tau/2} \int \frac{\eta(\mathbf{k}_B,h)(1+\cos^2\theta_B)G_r(\theta_x,\theta_y)G_t(\theta_x,\theta_y)}{R_i^2(R_s)^2} d\Omega dR \qquad (3)$$

where $G_r$ is the receiver antenna pattern, $R_0 = R_s + R_i$, $\theta_x, \theta_y$ are the direction cosines with respect to the receiver, $c$ the speed of light, and $\tau$ the pulse width. Note that we are assuming that $\eta$ has only a dependence on Bragg vector ($\mathbf{k}_B$) and altitude ($h$),
which is suitable for PMSE. In addition, we assume that the transmitted pulse and the receiver bandwidth have perfect square shapes.





The main characteristics of the monostatic and bistatic observations over MAARSY are summarized in Table 1. Note that the expected difference in sensitivity between monostatic and bistatic, assuming isotropic scattering, volume filling and considering range differences, is $\sim 26.9$ dB.

## 4 Experimental Results

5   In this section we present the results of the monostatic and bistatic observations conducted with MAARSY and KAIRA on August 12, 2016. In addition, we show the parameters that result from combining both systems.

### 4.1 MAARSY Monostatic Observations

Figure 4 shows the spectral parameters of five and a half hours of observations during this campaign: (a) Signal-to-noise ratio (SNR) in dB scale, (b) mean Doppler shift (Hz), and (c) spectral width. The spectra have been obtained with 1024 FFT points 10  and 32 coherent integrations. In all three measurements, we show only values satisfying an SNR greater than -6 dB. Each range profile is obtained every 40 seconds. The Doppler shifts vary between $\pm 3$ m/s with periods of a few minutes. The SNR shows a variety of strong and weak and wide and narrow layers around 85 km. After 0800 UT clearly the echoing region gets wider and at least three narrow layers are observed. The spectral widths show relatively low values with a median of 0.3 Hz and with little variability in both time and altitude, except for larger values for the layer around 75 km at 0430 UT, and the layers above 15  85 km between 0630 an 0700 UT.

  In general these PMSE observations are typical of monostatic systems, MAARSY being the most sensitive system able to measure echoes with the lowest RCSs. In this particular case, the estimated PMSE RCSs are between $1.0 \times 10^{-17}$ and $1.0 \times 10^{-11}$ m$^{-1}$. Recently Latteck and Strelnikova (2015) have reported observations of polar mesospheric echoes during all seasons and pointed out the type of echoes that were not observed previously with less sensitive systems, e.g., coexistence of 20  PMSE with lower mesospheric echoes around equinoxes.

  Another important feature in Figure 4a is the variability of SNR in both time and space. We will show later that such variability is mainly due to horizontal variability and not to in-situ temporal variability.

### 4.2 KAIRA Bistatic Observations

The corresponding KAIRA results are shown in Figure 5. The spectral parameters are similar to those shown for MAARSY in 25  Figure 4, but instead of altitude they are shown as function of total range. This time they were obtained every 20 seconds, and without any coherent integration. The spectra have been estimated using 1000 FFT points and 10 incoherent integrations. In the case of KAIRA we have used two conservative criteria to select the data, i.e., an SNR threshold of -10 dB and a coherence threshold of 0.25. By coherence we mean the coherence between the signals in both linear polarizations without subtracting the noise in the denominator. As a reference, we are plotting the corresponding height on the right, assuming that all the echoes are 30  observed over MAARSY. Clearly echoes below 290 km in range do not come from regions over MAARSY since they would have come from much lower heights.





To our surprise, we were able to observe echoes from ranges that do not correspond to PMSE echoes illuminated overhead MAARSY, i.e., at ranges closer than 290 km. The echoes, apparently originating from heights lower than the PMSE heights, are clearly connected to the strongest echoes which are located at the true PMSE heights. The only plausible explanation is that these echoes are normal PMSE echoes at normal PMSE altitudes, which are illuminated by the sidelobes of the MAARSY

transmitter beam and originate from a larger geographic area. This time the echoes are clearly observed to vary with time both in duration and in intensity. Moreover, in the case of Doppler shift, it is mainly negative varying with time and range. In the case of range, there is a predominant dependence, being smaller at closer ranges. As in the case of MAARSY, the spectral width are relatively small over MAARSY (total range farther than 290) and vary significantly at closer ranges, particularly after 0730 UT.

Around 0540 UT we are plotting a black parabolic line over the observed KAIRA PMSE (pointed by the black arrow). This line has been obtained assuming that a scattering center was originally located at 85 km in altitude at the middle point between KAIRA and MAARSY and drifted horizontally at a constant velocity. The velocity used is 68 m/s (from KAIRA to MAARSY), which is obtained from the Doppler measurements (see below). The agreement between the observed PMSE range-time behavior and this simple model is excellent, implying that the PMSE structures are drifting with the background

horizontal wind.

### 4.3 Combined KAIRA-MAARSY PMSE measurements

Now we combine both measurements in this section. The peak values after a 3-point smoothing of the monostatic (MAARSY) and bistatic (KAIRA) data, obtained from the same volume (overhead MAARSY) are shown in Figure 6a, in red and green, respectively. The horizontal velocity over MAARSY (in the direction KAIRA-MAARSY, being positive towards KAIRA),

also overhead MAARSY, is shown in blue (right axis). The horizontal velocity component in the direction MAARSY-KAIRA has been obtained from KAIRA's Doppler shift (Figure 5b) and MAARSY's vertical velocity (Figure 4b).

We can see that in general there is a good correspondence between the two SNR time-range variations, particularly when MAARSY signals are strong. To observe this feature better, in Figure 6b we plot MAARSY vs KAIRA peak values. In this plot we can identify an approximate difference in signal between the two of 30 dB, which we have marked with a vertical dashed

line.

Given the spectral widths shown in Figure 6c are almost constant, we present a 2-D histogram of MAARSY's SNR and spectral widths with the counts in log scale. The great majority of echoes have a strong variability in SNR with small changes in spectral width. A smaller but detectable population is characterized by an SNR that increases with increasing spectral width. Taking into the account the PMSE simulations shown in Figure 1b, we superimpose lines over these two populations

and labeling them as "ice-dominated" (blue) and "turbulence-dominated" (black), respectively. One can argue that the part of Figure 6c, where spectral width (proxy for turbulence intensity) increases with increasing RCS, agrees with the part of Figure 1b for small $S_c$ (black-dashed line with $S_c = 3$). The other part of Figure 1, that covers the majority population, where narrow spectral widths can produce any value of RCS, seems to agree with the horizontal line of Figure 1b. From this simple plot, we





assume in the remaining of the paper that most of KAIRA detections come from scattering regions with high Schmidt numbers. We are again marking the SNR threshold of 30 dB identified before.

Note that the spectral widths have not been corrected by any effects, like beam or shear broadening. Therefore, these values represent upper values of atmospheric turbulence intensity.

Having defined an empirical RCS difference between KAIRA bistatic and MAARSY monostatic of ∼30 dB, in Figure 7 we showed the parameters resulting from combining both observations: (a) vertical structure and (b) vertical velocity after using a MAARSY SNR threshold of 30 dB, (c) vertical structure over MAARSY as observed with KAIRA, and (d) inferred horizontal velocity from KAIRA and MAARSY Doppler velocities. The thresholded MAARSY SNR results show that the echoes come from a narrow region in altitude, appearing and disappearing in time. After 0700 UT a second narrow region appears at lower

altitudes, with larger RCS. The corresponding vertical velocity does not show a distinct altitude dependence. In the case of KAIRA, the observed structures are wide in range, as expected, due to the convolution of a narrow layer with a wide receiver beam.

In the case of the horizontal velocity, the estimates are consistent when a single drifting structure occurs, e.g., between 0515 and 0715 UT. When more structures occur simultaneously, the estimated horizontal velocity gets more complicated, e.g., at

total ranges smaller than 280 km and times around 0430 and 0800 UT. Assuming that PMSE structures over MAARSY have horizontally drifted with the obtained horizontal velocities, we have indicated 100 km segments in Figure 7a with white lines, namely for faster flows the segments are shorter in time, e.g., around 0600 UT.

## 5   Discussion

We start our discussion by arguing that KAIRA observations come from scattering regions with high Schmidt numbers. By

looking at the PMSE simulations, a wide range of RCSs (more than 6 orders of magnitude) for a constant turbulence intensity can be obtained by varying $S_c$ (see Figure 1b). In the 5 hours presented, MAARSY's observed PMSE SNR show a variability of more than 50 dB (i.e., more than 5 orders of magnitude in RCS). Such variability cannot be attributed to changes in other parameters, e.g., electron density, atmospheric viscosity, ice density, density gradients, etc. But they can be easily obtained by having coexistent ice particles with different radii ($r_A$) and therefore generating different $S_c$, i.e., $r_A = \sqrt{S_c/6.5}$ (e.g., Cho

et al., 1992; Rapp and Lübken, 2003). Therefore, given that KAIRA observations correspond to MAARSY SNR greater than ∼ 30 dB, i.e., $\eta > 5 \times 10^{-14}$ m$^{-1}$, we claim that such common observations arise from high $S_c$. Previous multi-wavelength studies have been also focused on PMSE with high $S_c$ (e.g., Hoppe et al., 1990; Belova et al., 2007; Naesheim et al., 2008; Rapp et al., 2008; Li et al., 2010; Li and Rapp, 2011).

High $S_c$ means that the observations occur in the viscous-convective subrange and therefore the PMSE RCS will have a $k_B^{-3}$

(or $\lambda_B^3$) dependence at the Bragg wavelengths of interest, i.e., $\lambda_B < 3$ m. In other words, we are ruling out a higher dependence on $\lambda_B$ at the wavelengths of interest, since such dependence will require small $S_c$ (see Figure 1a). Our PMSE measurements fall in the region between the green and red continuous lines in Figure 1a, i.e., covering $S_c$ between 100 and 900 within a





narrow region of RCS that spans about half an order of magnitude. The red curve ($S_c$=900) is still in the power law regime, while the green curve ($S_c$=100) is going down into the exponential regime.

Following these considerations, we now discuss the results related to PMSE drifting structures and to the PMSE angular dependence, separately. In addition, we briefly discuss other observed features and future plans.

## 5.1   PMSE Drifting Structures

The half-parabolic structures seen in Figure 5a are typical signatures of horizontally drifting structures. We have verified that this is the case by overlaying the expected trajectory (total range vs time) of a structure drifting at a constant velocity at 85 km altitude, and the match is perfect. To continue the analogy of a typical drifting echo, e.g., airplanes, the left half of the parabolic signature is not observed given that the KAIRA receiver beam points towards MAARSY (see Figure 3b). Therefore the left structures, i.e., between KAIRA and the middle point, are below KAIRA's sensitivity.

The horizontal distance between the middle point and MAARSY at 85 km is ∼90 km. Our results also show that these PMSE structures with high $S_c$ have a limited volume of approximately 5-15 km of horizontal extent in the KAIRA-MAARSY direction. Moreover, these PMSE "clouds" (limited-volume structures) present horizontal separations ranging from 20 to 60 km. These approximate distances and sizes have been obtained from Figure 7a. It is important to stress, that the horizontal structures we have identified are for PMSE with high $S_c$. A more sensitive bi-static radar configuration would have observed PMSE all the time, i.e., without spatial gaps, but with varying RCSs.

The obtained horizontal and vertical features of PMSE with high $S_c$ are consistent with NLC structures observed with lidar, airglow imagers, and ground-based NLC photography. For example, using lidars the NLC half-power full-width in height is approximately 1 km (e.g., Fiedler et al., 2009). Drifting NLC bright clouds with horizontal separations between 10 and 40 km are also typical of NLC observations (e.g., Baumgarten and Fritts, 2014, Figure 2). A sketch of what we are observing can already be found in Figure 2. PMSE clouds drift from the middle point between KAIRA and MAARSY to MAARSY. These clouds are of different sizes and have different separations. KAIRA is only able to observe the red clouds (PMSE with high $S_c$), while MAARSY monostatic can observed also the blue clouds (PMSE with lower $S_c$). So, the empty regions observed by KAIRA are not really empty, they are filled by blue clouds that are 'invisible' to KAIRA.

NLC observations with high resolution cameras from the ground imply that even horizontal features with smaller scales should be measured by radar, less than 1 km, and even at meter scales. Such scales are not possible in the current configuration. Recently, Sommer and Chau (2016) have been able to identify horizontal structures with sizes around 1 km using radar imaging and antenna compression techniques. We are planning to improve this resolution by using a combination of compressed sensing (Harding and Milla, 2013) and multi-input multiple output (MIMO) techniques (Urco et al., 2018).

The drifting nature of PMSE structures have been hypothesized before and sometimes characterized by simultaneous measurements at sites separated by 100-150 kms (e.g., Bremer et al., 1996; Rapp et al., 2008). Our measurements are the first to show directly such drifting PMSE structure with high $S_c$ as well as the limited-volume horizontal sizes and separations of few tens of kilometers between them.



Although our results are encouraging to further understand the temporal and spatial characteristics of PMSE and their relation to atmospheric dynamics and chemistry responsible of such characteristics, more detailed observations are needed. In our particular case, the whole PMSE region has experienced almost the same horizontal wind with a strong component in the KAIRA-MAARSY direction. This is not necessarily always the case, sometimes strong wind shears are observed. For

example using an EISCAT VHF tristatic experiment, Mann et al. (2016) observed that the upper part of PMSE moved in an opposite direction than the lower part. In that case, our observations would have shown the left/right part of the parabola for structures going towards KAIRA/MAARSY, assuming the limited-volume structures are maintained and move primarily in the KAIRA-MAARSY direction.

## 5.2   PMSE Angular and Wavelength Dependence

A by-product of our observations is the possibility of studying the angular dependence of PMSE. As indicated in Figure 3f, our bistatic configuration allow measurements with zenith angles ranging from $0°$ (middle point) to $\sim 33°$ (over MAARSY). To follow the previous literature, here we also characterized the angular dependence as follows

$$\eta(\theta) \propto \exp\left(-\frac{\sin^2\theta}{2\sin^2\theta_S}\right) \tag{4}$$

Before 2014, we would not have expected to observed these echoes given the high aspect sensitivity values reported in

the literature, i.e., $\theta_S = 2-3°$, particularly when so-called spaced antenna methods were employed (e.g. Zecha et al., 2001; Smirnova et al., 2012). Using a combination of vertical and oblique beams, the resulting values vary significantly, i.e., $\theta_S = 5-15°$ (e.g., Czechowsky et al., 1988; Huaman and Balsley, 1998; Zecha et al., 2001). Sommer et al. (2016) using many months of MAARSY multi-beam data, have hypothesized that PMSE are statistically due to localized isotropic scattering structures. Moreover, their hypothesis has been supported by Sommer and Chau (2016) who observed PMSE structures with

sizes around 1 km, i.e., smaller than the illuminated volume. Encouraged by the latter observations, we decided to add the MAARSY PMSE observations to the originally planned MMARIA campaign with KAIRA, i.e., the results we present here.

To determine the angular dependence, first we calculate the expected angular dependence assuming isotropic scattering. Moreover, we are assuming a narrow layer in altitude, centered at 85 km with a Gaussian width of 1 km. The expected received power is obtained after numerically integrating EQ 3, taking into account a range sampling of 300 m and $k_B^{-3}$ dependence in $\eta$.

We have simulated two scenarios: (a) assuming ideal antenna patterns like those shown in Figure 3 (Model 1), and (b) using a MAARSY antenna pattern with a random uniformly distributed amplitude varying from 0.2 to 1 in all 433 elements (Model 2). Model 2 simulates an extreme case of an antenna array with unmatched antenna elements. Measuring the actual power levels of the antenna sidelobes is not an easy task. In both cases the range dependence has been already considered.

In Figure 8a we show the resulting normalized received power as a function of total range for Model 1 and Model 2 with

black solid and dashed lines respectively. The main difference between the two is the expected received power arising from the sidelobes, i.e., the closer ranges. We compare the model profiles with three power profiles obtained from measurements: Measurement 1, along the black curve shown in Figure 5a; Measurement 2, average power between 0640 and 0725 UT; and Measurements 3, average power between 0730 and 0830 UT. The resulting measurement profiles are shown in red, green



and blue respectively. In all three cases, the profiles have been self-normalized to their peak value which corresponds to measurements over MAARSY.

Assuming that the same scattering center drifts from KAIRA to MAARSY and remains unchanged during this time (a few tens of minutes), we proceed to analyze their angular dependence by comparing the measurements to the model outputs. The

received power ratios, measurements over models, are shown in Figure 8b, with Model 1 in solid lines and with Model 2 in dashed lines. The color represents, as before, which measurements were used. On top of the ratios with Model 1 we plot with dashed-dotted lines the fits to EQ 4. The resulting $\theta_s$ values are 12.10°, 12.80°, and 13.19° for Measurements 1, 2, and 3, respectively. In the case of Model 2, $\theta_s$ is greater than 20°, however the obtain ratio profiles, do not behave like EQ 4.

Using the same procedure in Model 1 (ideal antenna gain) for a MAARSY-MAARSY configuration (monostatic), we found

that the expected power ratio of MAARSY measurements over KAIRA's measurements is 27 dB. Comparing this difference to the empirically determined difference of 30 dB, the difference in power with respect to isotropic scattering at 33° zenith angle is -3 dB, i.e., an equivalent $\theta_s \sim 27.55°$. In the case of MAARSY-MAARSY configuration using Model 2 (imperfect antenna gain), the expected ratio is 21.39 dB, i.e., a difference of 8.61 dB at 33° and $\theta_s \sim 15.87°$. In both cases, the actual values could be a few dB less, if the real antenna pattern of the KAIRA dipoles is included.

As one can see, we obtained different values of $\theta_s$ depending on what portion of the data we use, and which assumptions we make. From a simple inspection, Model 2 qualitatively has a better agreement with observations, i.e., power levels are almost constant at large zenith angles. However when compared to the angular dependence of EQ 4, the agreement is better with Model 1. We have assumed that the same PMSE structure drifts without changing much from KAIRA to MAARSY. Moreover, the PMSE clouds are elongated in the North-South direction and drift mainly in the zonal direction. In reality this

might not be the case, since we can not measure the horizontal velocity transverse to the KAIRA-MAARSY direction, and PMSE RCSs might have changed, spatially and temporarily, during the drifting process. Typically, correlation times of 2.8-m PMSE irregularities are in the order of seconds, while the observed kilometer-scale drifting structures appear to be frozen for a few tens of minutes. The deviation from our assumptions, i.e., the spatial and temporal evolution of PMSE RCS, might be the main reason for the lack of consistency of the angular dependence using different methodologies.

In general, the results are not perfectly consistent, i.e., we can not explain all the observations with the simple Gaussian model of EQ 4. However, we can conservatively conclude that our measurements indicate that by using the simple Gaussian model, the true $\theta_s$ for this event is greater than 12°, i.e., the scattering cannot be considered highly aspect sensitive. These results are in general consistent not only with previous multi-beam experiments but also with the suggestion by Sommer et al. (2016), i.e., PMSE scattering is in general not highly aspect sensitive as previously reported, but instead the scattering is due

to limited-volume (localized) isotropic structures.

The small differences between our estimates and the suggestion of Sommer et al. (2016), i.e., between slightly isotropic and isotropic, might be due to: (a) unknown behavior of the antennas at the sidelobe levels, and/or (b) selection of PMSE with high $S_c$. For the former, estimating the actual gain of sidelobes is not trivial given the mutual coupling of closely located neighboring antennas is hard to characterize. In the case of the latter, Sommer et al. (2016) included all PMSE measurements

during a month, with varying $S_c$, therefore the structures with larger $S_c$ could be less isotropic than structures with lower $S_c$.



### 5.2.1 Other features and Future Plans

In our results, we have used existing PMSE scattering theories, all of them showing a well defined $k_B^{-3}$ for the meter scale irregularities at high $S_c$ (viscous-convective subrange), and an exponential decay at smaller scales (viscous-diffusive subrange) (see EQ 1). In the viscous-diffusive subrange all theories show that RCS increases with turbulence intensity. However in the viscous-convective subrange, our simulations show that PMSE RCS decreases with increasing turbulence intensity (see Figure 1b) in the viscous-convective subrange. Such behavior is not reproducible using the expressions provided by Rapp et al. (2008) and Varney et al. (2011). Although difficult to validate observationally, unless the other parameters are measured (e.g., electron density, density and ice gradients, etc.), we think it is worth investigating such behavior, both theoretically and experimentally.

In Figure 6c, we show that two well-separated populations of polar mesospheric echoes in the summer based on their their SNR vs spectral width behavior, i.e., RCS vs turbulence intensity. The ice-dominated population belong to majority PMSE previously reported. The turbulence-dominated echoes correspond to: (a) echoes occurring below the typical PMSE altitudes, in our case around 75 km, and (b) echoes occurring at the top of PMSE, presumably with small particles sizes, i.e., small $S_c$. A similar behavior has been shown using EISCAT 224 MHz by Rapp and Hoppe (2006). In the case of the echoes occurring around 75 km, strictly speaking they might not be called PMSE, however their existence, besides enhanced turbulence, might require a way to reduce their diffusion time. The altitude is too low for ice, though. Its existence might be related to some of the mesospheric echoes observed at equatorial latitudes (e.g., Lehmacher et al., 2009) and polar mesospheric echoes observed in winter (e.g., Latteck and Strelnikova, 2015) that can not be explained by pure turbulence arguments.

In future experiments, we plan to improve the measurements by focusing on PMSE and making better use of MAARSY and KAIRA capabilities. For example, we plan to steer MAARSY in different directions towards KAIRA and generate also different KAIRA beams towards MAARSY simultaneously. In this way, we would improve the quality of the observations, the spatial coverage, and the angular dependence. This type of observations would be a good complement to current NLC studies from the ground, since they can be done independent of weather conditions as long as there are some electrons and sufficient ice particles with relative large radius (i.e., high $S_c$). Moreover, the MAARSY observations close to overhead bring the additional advantage that it allows the observations of echoes due to ice particles with smaller sizes than those responsible of NLCs. Besides the improved PMSE characteristics, the proposed improved experiments would allow also wind field measurements around the summer polar mesopause with unprecedented temporal and spatial resolutions. Instead of specular meteor echoes, one could applied the MMARIA approach (e.g., Stober and Chau, 2015; Chau et al., 2017) to PMSE observations at multiple locations and from different observing angles.

## 6 Conclusions

We show for the first time direct evidence of PMSE limited-volume structures drifting with the background atmospheric wind using a combination of monostatic and bistatic observations. The observed structures have horizontal sizes between 10 and 20 km, separations between 20 and 60 km, and vertical widths of less than 1 km. These features have been observed on PMSE echoes with high $S_c$, and are consistent with previously reported features of NLCs.



We have also investigated the angular dependence of these PMSEs with high $S_c$. We find that during this event, PMSE scattering is in general not highly aspect sensitive. A conservative lower bound estimate for the aspect sensitivity parameter is $\theta_s = 12°$. Depending on the measurements and assumptions we make, the estimate gets closer to isotropic. Our results are consistent with most recent works on PMSE angular sensitivity that indicate that PMSE scattering is mainly composed of

5    localized (limited-volume) isotropic structures.

Improved experiments using the full beam steering and beam forming capabilities of MAARSY and KAIRA, respectively, will be helpful to (a) resolve better the PMSE spatial and temporal characteristics at 10-100 km scales, (b) improve the angular dependence of PMSE, at least at large $S_c$. Besides studying the spatial-temporal features of PMSE, the results of improved experiments could be used to get wind field estimates with unprecedented spatial and temporal resolutions using PMSE as

10   tracers in an MMARIA approach, which utilizes a multi-static network of low power all-sky illuminating radars.

*Data availability.* The MAARSY and KAIRA spectra data are available. Interested users, please contact the main author for the MAARSY and KAIRA spectral data. MAARSY raw data should be requested to R. Latteck.

*Author contributions.* TEXT

*Competing interests.* TEXT

15   *Disclaimer.* TEXT

*Acknowledgements.* We would like to thank Irina Strelnikova for her suggestions for improvements and for providing suitable references. We would also thank Kiara Chau for preparing the MAARSY-KAIRA sketch. This work was partially supported by the WATILA Project (SAW-2015-IAP-1). KAIRA was funded by the University of Oulu and the FP7 European Regional Development Fund and is operated by Sodankylä Geophysical Observatory with assistance from the University of Tromsø.



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





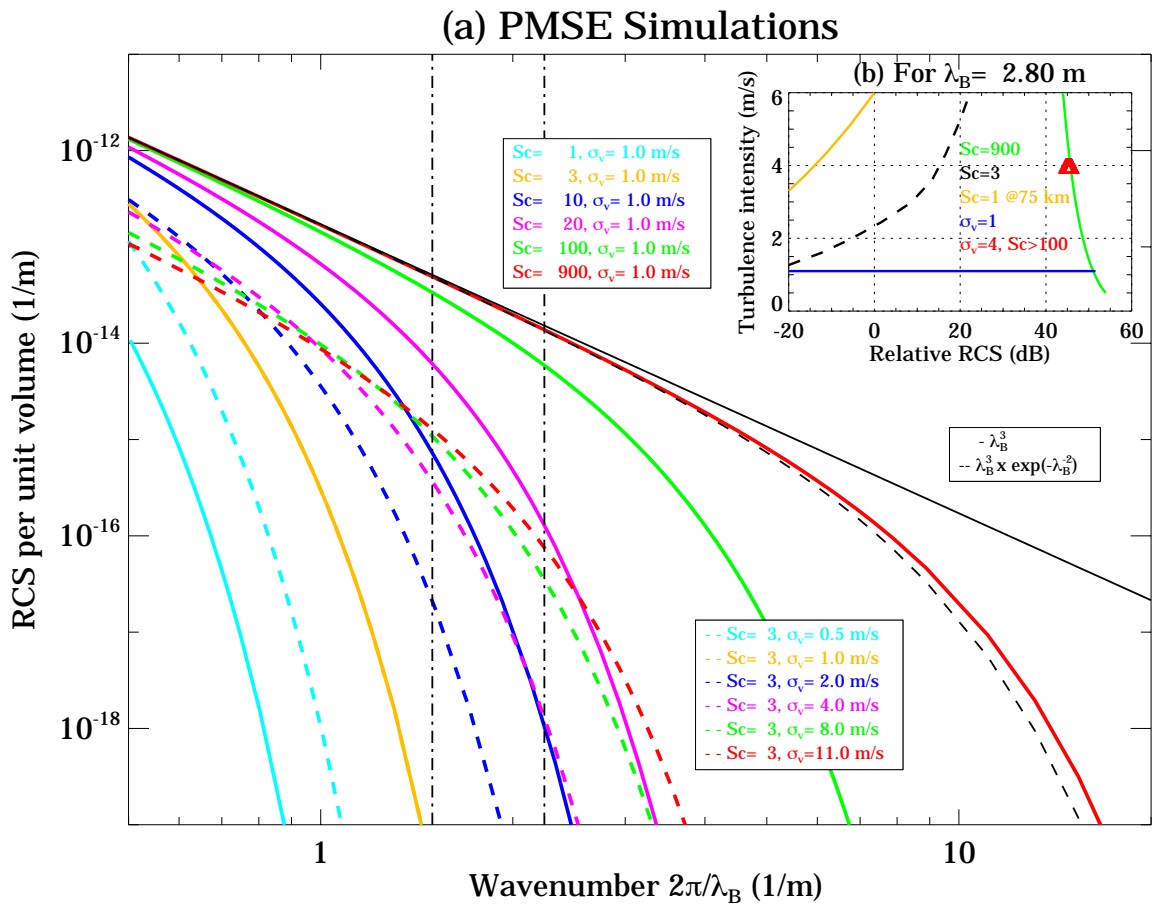

**Figure 1.** Simulations of PMSE RCS as a function of (a) wavenumber and (b) turbulence intensity at a $\lambda_B = 2.8$ m. In (a) we show two simulations, one keeping $S_c = 3$ constant and varying $\sigma_v$, and the second one keeping $\sigma_v = 1.1$ constant and varying $S_c$. Two approximate dependence on $\lambda_B$ are shown in black (Varney et al., 2011, Eq. 44). In the case of (b), five cases are shown (see text for details). The two vertical dashed lines represent the wavenumbers of interest, i.e., bistatic middle point (smaller) and MAARSY monostatic (larger).




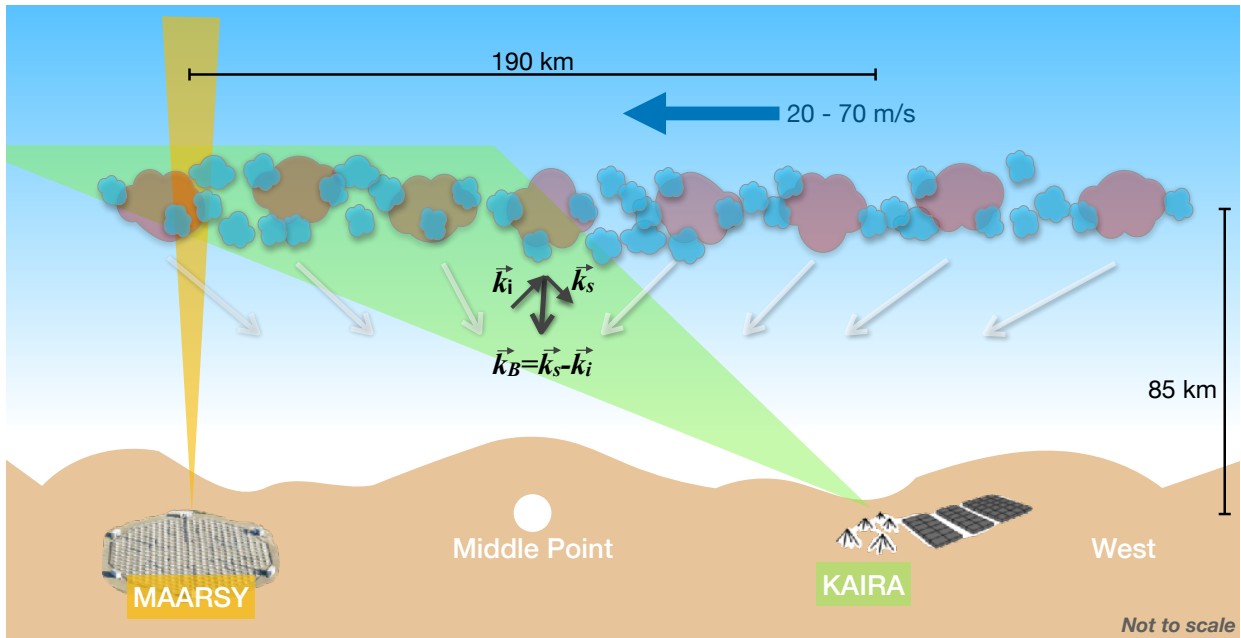

**Figure 2.** Sketch of PMSE observations with MAARSY as transmitter and KAIRA and MAARSY as receivers. The region of PMSE are depicted with clouds of different sizes and colors located in a narrow region. The white arrows represent the expected Bragg vectors of the MAARSY-KAIRA detections (see text for more details).

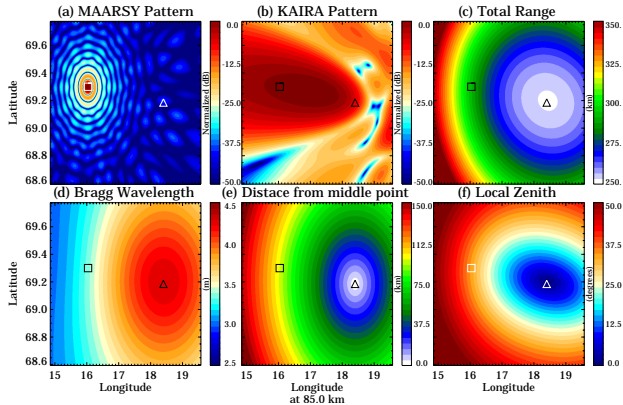

**Figure 3.** Antenna patterns and geometric parameters for the MAARSY-KAIRA multistatic configuration as function of longitude and latitude at 85 km. (a) MAARSY one way transmitting patter, (b) KAIRA narrow beam pointing over MAARSY, (c) total range, (d) Bragg wavelength, (e) distance with respect to the middle point, and (f) the resulting zenith angle with respect to local zenith. The locations of MAARSY and the middle point are indicated with a square and triangle symbols, respectively.





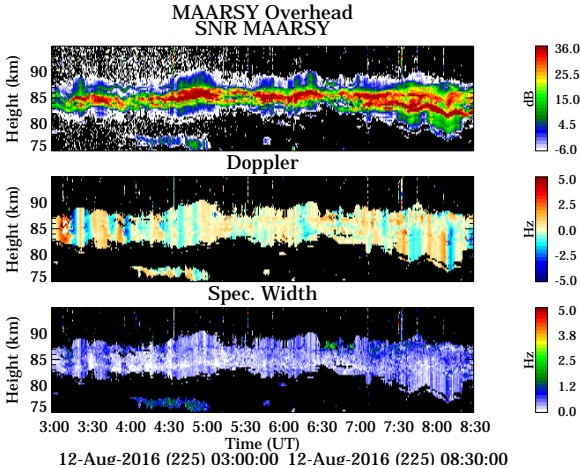

**Figure 4.** PMSE height-time observations using MAARSY for transmission and reception on August 12, 2016: (a) Signal-to-noise ratio (SNR), (b) Doppler shift, and (c) spectral width.

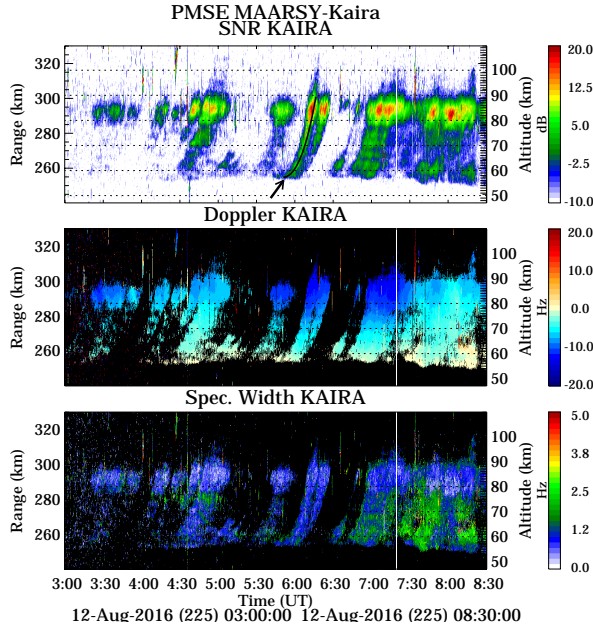

**Figure 5.** PMSE range-time observations using MAARSY for transmission and KAIRA for reception on August 12, 2016: (a) Signal-to-noise ratio (SNR), (b) Doppler shift, and (c) spectral width. The approximate height is indicated on the right, assuming the strongest echoes are observed over MAARSY. In (a) a black line is plotted around 0600 UT over a drifting structure (pointed by a black arrow).





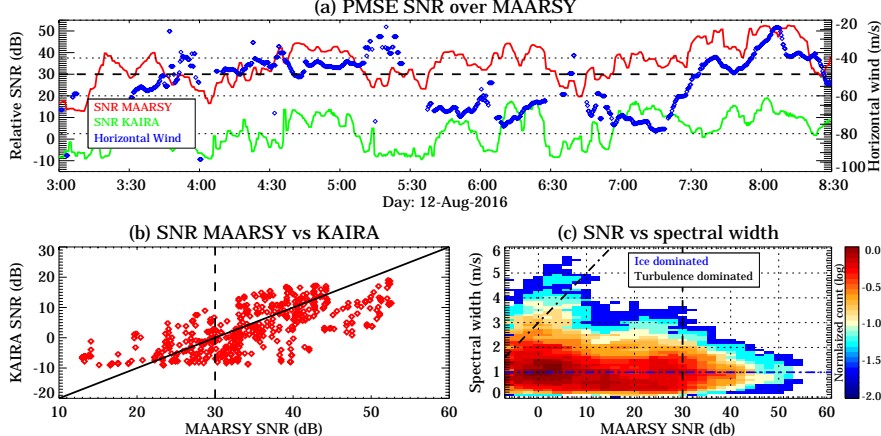

**Figure 6.** Combined MAARSY and KAIRA measurements: (a) Median SNR over MAARSY as observed with MAARSY and KAIRA, and over the middle point; (b) MAARSY vs KAIRA SNR scatter plot; and (c) bivariate distribution of MAARSY SNR vs spectral width. The dashed lines indicate MAARSY's 30 dB SNR as a reference. In (a), the mean horizontal wind over MAARSY in the direction MAARSY-KAIRA is shown in blue.

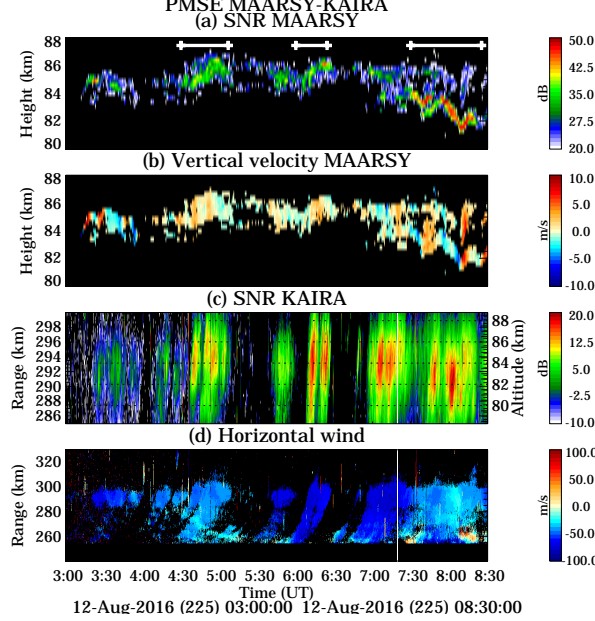

**Figure 7.** Derived PMSE parameters from combining MAARSY and KAIRA observations, using a SNR threshold of 30 dB in MAARSY observations: (a) Vertical structure, (b) vertical velocity, (c) vertical structure as observed from KAIRA, and (d) horizontal velocity along MAARSY-KAIRA direction (positive towards KAIRA). Approximate distances of 100 km are indicated in (a) with white lines. See text for details.




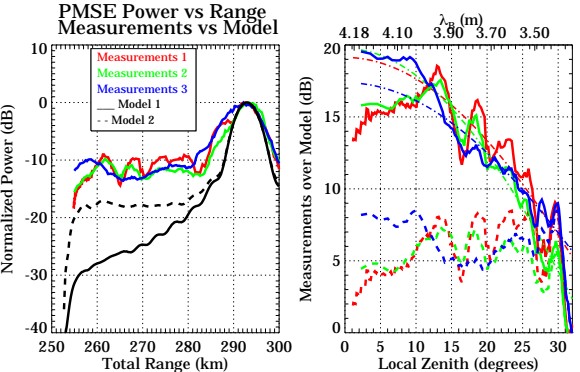

**Figure 8.** (left) Normalized power cuts as function of total range for three different time periods in Figure 5a in red, green, and blue The expected relative power for an isotropic scattering is indicated in black. (right) Power ratio between measurements and isotropic power. Fitted curves centered at 0 degrees are indicated in dashed-dotted lines.

**Table 1.** Experimental Parameters over MAARSY

| Parameter | MAARSY-MAARSY | MAARSY-KAIRA |
|---|---|---|
| Geometry | Monostatic | Bistatic |
| Transmitter Elements | 433 | 433 |
| Receiving Elements | 433 | 48 |
| Peak Power | 800 kW | 800 kW |
| $\lambda_B$ | 2.8 m | 3.3 m |
| PMSE mean range | 85 km | 294 km |
| Received Power relative to Monostatic | 0 dB | -26.90 dB |