# Peer review of "Multi-static spatial and angular studies of polar mesospheric summer echoes combining MAARSY and KAIRA"

_Atmospheric Chemistry and Physics, 2018_

## Referee Comment (RC1) · Anonymous Referee #2 · 25 Apr 2018

The authors present PMSE measurements obtained by a unique setup, a combination of vertical monostatic sender/receiver and a receiver tilted towards the primary system with 180 km baseline. They surprisingly observed PMSE above the middle point illuminated by the sidelobes of the primary system. With a valid assumption on PMSE altitude they were able to register the horizontal movement of PMSE structures. They observed drifting structures and the estimated horizontal scales correspond to scales known from NLC observations. The special setup allowed to constrain the lower limit of the angular sensitivity disproving that PMSE are highly aspect sensitive. The measurement results are carefully interpreted related to PMSE scattering theories.

[Figure]

The paper is very well written and explained. Efforts were made to investigate this unique case study as comprehensive as possible. Clearly, the results of the drifting PMSE structures are intriguing, and corresponds well with the expectations derived from NLC observations. Because this is the first experimental evidence for horizontally drifting PMSE structures and it demonstrates the high potential of this kind of measurements, this work is of high scientific value and suitable for publication in ACP. I have only minor remarks, mostly language, which are listed below.

p. 1, l. 1: Noctilucent clouds -> noctilucent clouds

p. 1, l. 2: the 3 m Bragg wavelenth refers to PMSE only, not to NLC, this part could be reworded to make this clearer

p. 1, l. 6: have horizontal widths

p. 1, l. 15: over high (or polar) latitudes

p. 2, l. 10: you show later that the area illuminated is much wider than these mentioned few km

p. 2, l. 20: during special atmospheric conditions

p. 2, l. 20: special in what way?

p. 4, l. 7: 2.8 m

p. 4, l. 7: is Sc0 -> Sc?

p. 4, l. 8: fix -> fixed

p. 4, l. 21: delete for in "and for k_B"

p. 4, l. 32. thae -> the

p. 5, l. 1-2: these two sentences are not consistent. Either this configuration allows only one Bragg vector or multiple

p. 5, l. 12: Could you explain more clearly about the horizontal width of the MAARSY reception beam, in relation to Fig. 3a? Does MAARSY also receive (a minor partition of) power from the side lobes above the middle point as well? And could MAARSY be configured to steer a single, localized reception beam towards the middle point? The horizontal extent using the imaging approach by Sommer and Chau (2016) is limited to +-15 km, so maybe not.

p. 6, l. 12 "a horizontal distance" -> "horizontal distance with respect to middle point"

p. 6, l. 19: and its located -> and is located

p. 7, l. 27: an SNR -> a SNR

p. 8, l. 7: width -> widths

p. 8, l. 8: 290 -> 290 km

p. 8, l. 20: delete "also overhead MAARSY", it's mentioned before the brackets already

p. 8, l. 29: Taking into account

p. 8, l. 30 and label them as

p. 9, l. 6: we show the parameters

p. 10, l. 17: "with NLC structures as known from ..." Otherwise this sentence can be misunderstood as if you had these additional data for this date and location

p. 10, l. 23: "while MAARSY monostatic ..." please check grammar of this sentence

p. 10, l. 25: please add the citation here as well

p. 10, l. 26: can you provide an estimate of your limits?

p. 10, l. 32: structure -> structures

p. 11, l. 11: allows for measurements

p. 11, l. 14: to observe

p. 12, l. 8: the obtained ratio profiles

p. 12, l. 34: In case of

p. 13, l.6: remove "in the viscous-convective subrange", it's double

p. 13, l. 9: delete double "their"

p. 13, l. 10: belongs to

p. 13, l. 10: reword "majority PMSE"

p. 13, l. 11: remove "that" in "we show that two"

p. 13, l. 11: "polar mesospheric echoes in the summer" -> PMSE

p. 13, l. 24: allows for the observations

p. 13, l. 25: would also allow

Fig. 3: (e) Distace -> Distance

Fig. 8: Caption: dot between "blue The expected"

---

## Referee Comment (RC2) · Anonymous Referee #3 · 26 Apr 2018

This is a very interesting manuscript. Although I am not familiar with the literature dealing with the use of KAIRA in conjunction with the EISCAT systems, this is the first paper I am aware of describing its use in conjunction with an MST radar. This leads to results that could not be obtained from an MST radar operating in isolation.

I have no fundamental problems with the scientific content of this manuscript. However, there are a large number of places where I was not sure what the authors were trying to say or thought that their ideas could have been expressed more clearly. These are indicated below. I do not expect the corrections to signficantly change my view of the manuscript.

[Figure]

Note that there is typically a mismatch between the indicated line numbers and the actual ones. I have tended to use the actual line numbers for parts of the manuscript that appear at the top of the page, but the indicated ones for parts lower down.

- page 3, line 17. I presume that the symbol nu in the formula for Schmidt number should have a subscript a? For completeness, the units for each of the parameters involved in equation 1 should be stated here. I realise that these are given at the bottom of page 3 when specified values are quoted.

- In Figure 1, the value of RCS is shown along in the y axis in the main plot, but along the x axis in the inset plot (1b). It would be more consistent if these values were shown along the same axis in both cases.

- page 3, line 25. It would be better to use the words "lowest and largest" rather than "lower and larger" in the following sentence: "The vertical dashed-dot-dashed lines represent the lower and larger . . ."

- page 4, line 5 and Figure 1. The units for sigma_v (presumably m s-1?) should be shown for completeness. As a more general point, Doppler shifts and spectral widths are sometimes shown in units of Hz (e.g. Figure 4) and sometimes in units of m s-1. It would be better to use m s-1 units throughout.

- page 4, lines 5 - 15. Points 1 and 4 both refer to high values of Sc, but are separated by points about moderate (2) and low values (3). This summary would read more clearly if points 1 and 4 were shown adjacent to each other.

- Figure 2. I initially found this figure confusing with the the Bragg wavenumbers shown at the mid-point between KAIRA and MAARSY since MAARSY is being operated with a vertical beam, i.e. with k_i vertically directed. It is only later in the manuscript, when the idea of MAARSY sidelobes is introduced, that this makes sense. It would be useful to make some forward reference to this when Figure 2 is first mentioned (page 4) so that the reader understands why it is shown as it is.

[Figure]

- page 6, line 25. The symbol G_r is described as the receiver antenna pattern whereas G_t (line 18) is described as the transmitter antenna gain. I realise that the term gain implies antenna transmit/receive pattern, but it would better to stick to the word gain for consistency.

- page 7 line 1: "Recently Latteck and Strelnikova (2015) have reported observations of polar mesospheric echoes during all seasons and pointed out the type of echoes that were not observed previously with less sensitive systems, e.g. coexistence of PMSE with lower mesospheric echoes around equinoxes." Is this last part true? I would have thought that there is more than a month between the spring equinox and the first PMSEs and between the last PMSEs and the autumn equinox.

-Figure 7. It would be better to use the y-axis label "Total range" - rather than "range" to avoid confusion with horizontal - separation. I realise that this is stated at the bottom of page 7, - but it is not indicated in the figure caption.

- page 8, line 4: "This time the echoes are clearly observed to vary with time both in duration and intensity." I am not sure what the authors mean by "varying with time in duration".

- page 8, line 5. I think that the word "systematic" would be better than "predominant" in the following sentence: "In the case of range, there is a PREDOMINANT dependence."

- in the relation to figure 6, the authors should state at what total range/altitude the velocity and SNR data are taken. Presumably the 3 point smoothing is in time rather than altitude/total range?

- page 8, line 21. Do the authors really mean "time-range" or just "time" in the following sentence: "We can see that in general there is a good correspondence between the two SNR TIME-RANGE variations . . ."

- page 8, line 22. What does the following sentence mean: "To observe this feature better, in Figure 6b we plot MAARSY vs KAIRA peak values". Peak with respect to

what?

- page 8, line 23: "In this plot we can identify an approximate difference in signal between the two of 30 dB, which we have marked with a vertical dashed line." Surely this difference represents the intersect of the solid back line with the y-axis (or rather, where MAARSY SNR is equal to 0.0 dB). The dashed black line does not represent this.

- Figure 6b. It would be better to use the same lengths for the x and y axes since they both cover the same intervals between minimum and maximum values.

- page 8, line 25. "Given that the spectral widths shown in Figure 6c are almost constant". I would say that that the spectral widths cover a large range, so I am not sure what the authors were intending to say here.

- page 8 line 26. "The great majority of echoes have a strong variability in SNR with small changes in spectral width." I understand the point that the authors are trying to make here, in defining a population. However, SNRs and spectral widths are very different things and so their values cannot be compared simply.

- Figure 7. Why have different ranges of range been used for the y-axes in panels c and d? It would make more sense to use the same.

- page 9 line 5. "Having defined an empirical RCS difference between the KAIRA bistatic and MAARSY monostatic of $\sim$ 30 dB . . ." It would be more consistent to refer to this as an SNR difference (as in Figure 6) rather than an RCS difference. I realise that one implies the other.

- page 9, line 11. "In the case of the horizontal velocity, the estimates are consistent when a single drifting structure occurs . . ." What exactly do the authors mean by this? That the observed velocity pattern is consistent with a structure moving at a single speed?

- page 10, line 9. "Our results also show that these PMSE structures with high Sc have

a limited volume of approximately 5 - 15 km of horizontal extent in the KAIRA-MAARSY direction." Could the authors explain in more detail how they infer this - and the "cloud" separations.

- page 10, line 20. "KAIRA is only able to observe the red clouds . . .". There is nothing in Figure 2 that I would describe as "red". If I understand the authors correctly, I would describe these structures are "light brown", "buff", or "beige".

---

## Author Comment (AC1) · 14 May 2018

We thank the reviewer for his/her encouraging comments and suggestions, that help us improve our paper. We will address and incorporate each of them in the revised version.

Regarding MAARSY pointing, the results away from MAARSY's zenith were were obtained from sidelobes. MAARSY could be pointed towards KAIRA but not quiet to the middle point. Join ting to the middle point would produce a second strong grating lobe in the opposite direction. In future experiment, we do plan to use different pointing directions in direction to KAIRA.

---

## Author Comment (AC2) · 14 May 2018

We thank the reviewer for the positive comments and suggestions that have helped us improve our paper. We will address them one by one in the revised version. Here we would like to comment about:

a) MAARSY-KAIRA diagram. We will add a clarifying text to indicate that the MAARSY illuminated volume is not limited to its zenith position, but other regions are also illuminated with the sidelobes, so the reader can understand the reason for the bistatic Bragg vector at the middle point.

[Figure]

b) Regarding the coexistence of lower mesospheric echoes (typically associated to winter echoes at polar regions) and the well0-known Polar mesospheric summer echoes (PMSE), in the revised version we will be more precise. The reviewer is wright, they do not co-exist at equinoxes, the lower mesospheric echoes co-exist with PMSE at the beginning of the PMSE season for many weeks, i.e., a month or so after the Spring equinox.

---

## Author Response (AR1)

May 24 , 2018

Dear Dr. Martin Dameris

Reference: # acp-2018-210  Response to the referees

Title: "Multi-static spatial and angular studies of polar mesospheric summer echoes combing MAARSY and KAIRA"

Dear Editor,

We thank the reviewers for appreciating our work and for helping us improve it. Below please find the specific answers and actions in *italics*. We are also including a marked up version of the revised file at the end.

Sincerely yours,

Prof. Dr. Jorge L. Chau
Head of the Radar Remote Sensing Department
Leibniz Institute of Atmospheric Physics at the Rostock University
chau@iap-kborn.de

Response to reviewer 2

The authors present PMSE measurements obtained by a unique setup, a combination of vertical monostatic sender/receiver and a receiver tilted towards the primary system with 180 km baseline. They surprisingly observed PMSE above the middle point illuminated by the sidelobes of the primary system. With a valid assumption on PMSE altitude they were able to register the horizontal movement of PMSE structures. They observed drifting structures and the estimated horizontal scales correspond to scales known from NLC observations. The special setup allowed to constrain the lower limit of the angular sensitivity disproving that PMSE are highly aspect sensitive. The measurement results are carefully interpreted related to PMSE scattering theories.

The paper is very well written and explained. Efforts were made to investigate this unique case study as comprehensive as possible. Clearly, the results of the drifting PMSE structures are intriguing, and corresponds well with the expectations derived from NLC observations. Because this is the first experimental evidence for horizontally drifting PMSE structures and it demonstrates the high potential of this kind of measurements, this work is of high scientific value and suitable for publication in ACP. I have only minor remarks, mostly language, which are listed below.

*R: We thank the reviewer for appreciating our work and for providing useful feedback to improve it.*

p. 1, l. 1: Noctilucent clouds -> noctilucent clouds
*R: Done.*

p. 1, l. 2: the 3 m Bragg wavelenth refers to PMSE only, not to NLC, this part could be reworded to make this clearer
*R: Reworded.*

p. 1, l. 6: have horizontal widths
*R: Done.*

p. 1, l. 15: over high (or polar) latitudes
*R: Done.*

p. 2, l. 10: you show later that the area illuminated is much wider than these mentioned few km
*R: We have added a clarifying text, that the mentioned few kms are related to the main beam.*

p. 2, l. 20: during special atmospheric conditions
*R: We have removed this part, since the special conditions are needed for our KAIRA measurements, namely that there is not a strong horizontal wind shear.*

p. 2, l. 20: special in what way?
*R: We have removed this sentence, see above.*

p. 4, l. 7: 2.8 m
*R: Done.*

p. 4, l. 7: is Sc0 -> Sc?
*R: Done.*

p. 4, l. 8: fix -> fixed
*R: Done.*

p. 4, l. 21: delete for in "and for k_B"
*R: Done.*

p. 4, l. 32. thae -> the
*R: Done.*

p. 5, l. 1-2: these two sentences are not consistent. Either this configuration allows only one Bragg vector or multiple.
*R: We have modified the sentence and add "and neglecting antenna sidelobes only one Bragg vector contributes …"*

p. 5, l. 12: Could you explain more clearly about the horizontal width of the MAARSY reception beam, in relation to Fig. 3a? Does MAARSY also receive (a minor partition of) power from the side lobes above the middle point as well? And could MAARSY be configured to steer a single, localized reception beam towards the middle point? The horizontal extent using the imaging approach by Sommer and Chau (2016) is limited to +-15 km, so maybe not.
*R: We have added a text related to the antenna sidelobes. If the PMSE is very strong, see the effects of the sidelobes in the main beam, however since MAARSY antenna pattern is used in transmission and reception, the two-way sidelobes is 2 times weaker (in dB) than the one way, making these effects noticeable under very strong PMSE conditions (more than 40 dB SNR). Regarding the pointing direction, narrow beams could be steered in direction to the middle point up to 30 degrees or so without the appearance of grating lobes in the opposite direction. In future experiments, we plan to add narrow beams towards KAIRA, even not perfect beams over the middle points to study in more detail the aspect sensitivity and the spatial and temporal characteristics of PMSE.*

p. 6, l. 12 "a horizontal distance" -> "horizontal distance with respect to middle point"
*R: Done.*

p. 6, l. 19: and its located -> and is located

*R: Done.*

p. 7, l. 27: an SNR -> a SNR
*R: Done.*

p. 8, l. 7: width -> widths
*R: Done.*

p. 8, l. 8: 290 -> 290 km
*R: Done.*

p. 8, l. 20: delete "also overhead MAARSY", it's mentioned before the brackets already
*R: Done.*

p. 8, l. 29: Taking into account
*R: Done.*

p. 8, l. 30 and label them as
*R: Done.*

p. 9, l. 6: we show the parameters
*R: Done.*

p. 10, l. 17: "with NLC structures as known from ..." Otherwise this sentence can be misunderstood as if you had these additional data for this date and location
*R: Done.*

p. 10, l. 23: "while MAARSY monostatic ..." please check grammar of this sentence
*R: We have rewritten the sentence, now reads "while MAARSY alone (i.e., monostatic) can observe"*

p. 10, l. 25: please add the citation here as well
*R: Done.*

p. 10, l. 26: can you provide an estimate of your limits?
*R: Done. We have added, that scales less than 1 km are not possible with the current configuration. This is mainly due to the receiving configuration used that is limited to the MAARSY antenna area. In an on-going effort using MIMO, we expect to improve the angular resolution by at least 50 %.*

p. 10, l. 32: structure -> structures
*R: Done.*

p. 11, l. 11: allows for measurements
*R: Done.*

p. 11, l. 14: to observe
*R: Done.*

p. 12, l. 8: the obtained ratio profiles
*R: Done.*

p. 12, l. 34: In case of
*R: Done.*

p. 13, l.6: remove "in the viscous-convective subrange", it's double
*R: Done.*

p. 13, l. 9: delete double "their"
*R: Done.*

p. 13, l. 10: belongs to
*R: Done.*

p. 13, l. 10: reword "majority PMSE"
*R: Done.  Now reads "The ice-dominated population belongs to the majority of PMSE events previously reported."*

p. 13, l. 11: remove "that" in "we show that two"
*R: Done.*

p. 13, l. 11: "polar mesospheric echoes in the summer" -> PMSE
*R: We have preferred to leave it like that, since we want to stress that not all the mesospheric echoes occurring at polar latitudes have the well-known characteristics of  PMSE, i.e., the need of ice-particles. For example the echoes around 77 km in Figure 4, are not part of what is known as PMSE.*

p. 13, l. 24: allows for the observations
*R: Done.*

p. 13, l. 25: would also allow
*R: Done.*

Fig. 3: (e) Distace -> Distance
*R: Done.*

Fig. 8: Caption: dot between "blue The expected"
*R: Done.*

Response to reviewer 3

This is a very interesting manuscript. Although I am not familiar with the literature dealing with the use of KAIRA in conjunction with the EISCAT systems, this is the first paper I am aware of describing its use in conjunction with an MST radar. This leads to results that could not be obtained from an MST radar operating in isolation. I have no fundamental problems with the scientific content of this manuscript. However, there are a large number of places where I was not sure what the authors were trying to say or thought that their ideas could have been expressed more clearly. These are indicated below. I do not expect the corrections to significantly change my view of the manuscript.
*R: We thank the reviewer for the encouraging comments and specific suggestions to improve our paper.*

Note that there is typically a mismatch between the indicated line numbers and the actual ones. I have tended to use the actual line numbers for parts of the manuscript that appear at the top of the page, but the indicated ones for parts lower down.

- page 3, line 17. I presume that the symbol nu in the formula for Schmidt number should have a subscript a? For completeness, the units for each of the parameters involved in equation 1 should be stated here. I realise that these are given at the bottom of page 3 when specified values are quoted.
*R: Done.*

- In Figure 1, the value of RCS is shown along in the y axis in the main plot, but along the x axis in the inset plot (1b). It would be more consistent if these values were shown along the same axis in both cases.
*R: We have preferred to leave as it is to be consistent with other works reporting the relationship of spectral width versus SNR (or RCS), e.g., Chau and Kudeki [2013], Patra et al. [2011].*

- page 3, line 25. It would be better to use the words "lowest and largest" rather than "lower and larger" in the following sentence: "The vertical dashed-dot-dashed lines represent the lower and larger . . ."
*R: Done.*

- page 4, line 5 and Figure 1. The units for sigma_v (presumably m s-1?) should be shown for completeness. As a more general point, Doppler shifts and spectral widths are sometimes shown in units of Hz (e.g. Figure 4) and sometimes in units of m s-1. It would be better to use m s-1 units throughout.
*R: We have added the units of sigma_v. Since we are dealing with monostatic and bistatic measurements, we have preferred to leave the units in Hz for Doppler shift. In case we convert to velocity, we use m/s and refer to it as Vertical velocity or Doppler Velocity instead of Doppler shift.*

- page 4, lines 5 - 15. Points 1 and 4 both refer to high values of Sc, but are separated by points about moderate (2) and low values (3). This summary would read more clearly if points 1 and 4 were shown adjacent to each other.
*R: Done.*

- Figure 2. I initially found this figure confusing with the the Bragg wavenumbers shown at the mid-point between KAIRA and MAARSY since MAARSY is being operated with a vertical beam, i.e. with k_i vertically directed. It is only later in the manuscript, when the idea of MAARSY sidelobes is introduced, that this makes sense. It would be useful to make some forward reference to this when Figure 2 is first mentioned (page 4) so that the reader understands why it is shown as it is.
*R: We have added in the sketch and description the MAARSY sidelobes and also indicate in the text to look for more details later.*

- page 6, line 25. The symbol G_r is described as the receiver antenna pattern whereas G_t (line 18) is described as the transmitter antenna gain. I realise that the term gain implies antenna transmit/receive pattern, but it would better to stick to the word gain for consistency.
*R: We are using now "pattern" in both cases, to stress the angular dependence.*

- page 7 line 1: "Recently Latteck and Strelnikova (2015) have reported observations of polar mesospheric echoes during all seasons and pointed out the type of echoes that were not observed previously with less sensitive systems, e.g. coexistence of PMSE with lower mesospheric echoes around equinoxes." Is this last part true? I would have thought that there is more than a month between the spring equinox and the first PMSEs and between the last PMSEs and the autumn equinox.
*R: We have corrected the text as follows "… coexistence of PMSE with lower mesospheric echoes for a few weeks at the beginning of PMSE"*

-Figure 7. It would be better to use the y-axis label "Total range" - rather than "range" to avoid confusion with horizontal - separation. I realise that this is stated at the bottom of page 7, - but it is not indicated in the figure caption.
*R: Done.*

- page 8, line 4: "This time the echoes are clearly observed to vary with time both in duration and intensity." I am not sure what the authors mean by "varying with time in duration".
*R: We have modified the sentence.*

- page 8, line 5. I think that the word "systematic" would be better than "predominant" in the following sentence: "In the case of range, there is a PREDOMINANT dependence."
*R: Done.*

- in the relation to figure 6, the authors should state at what total range/altitude the velocity and SNR data are taken. Presumably the 3 point smoothing is in time rather than altitude/total range?

*R: We have clarified the text, now reads "The peak values in range after a 3-point smoothing in time of the monostatic (MAARSY) and bistatic (KAIRA) data …" We have also added the ranges/altitudes used to obtain the peak values.*

- page 8, line 21. Do the authors really mean "time-range" or just "time" in the following sentence: "We can see that in general there is a good correspondence between the two SNR TIME-RANGE variations . . ."

*R: We meant just time, it is corrected now.*

- page 8, line 22. What does the following sentence mean: "To observe this feature better, in Figure 6b we plot MAARSY vs KAIRA peak values". Peak with respect to what?

*R: Peak values in the selected ranges/altitudes (see above).*

- page 8, line 23: "In this plot we can identify an approximate difference in signal between the two of 30 dB, which we have marked with a vertical dashed line." Surely this difference represents the intersect of the solid back line with the y-axis (or rather, where MAARSY SNR is equal to 0.0 dB). The dashed black line does not represent this.

*R: We have modified the text, now reads "…i.e., where KAIRA SNR is equal to zero which we have marked with a vertical dashed line"*

- Figure 6b. It would be better to use the same lengths for the x and y axes since they both cover the same intervals between minimum and maximum values.

*R: Done.*

- page 8, line 25. "Given that the spectral widths shown in Figure 6c are almost constant". I would say that that the spectral widths cover a large range, so I am not sure what the authors were intending to say here.

*R: We have modified the text to "Given the spectral widths 5  shown in Figure 6c show a weak dependence with respect to SNR for the majority of echoes, …"*

- page 8 line 26. "The great majority of echoes have a strong variability in SNR with small changes in spectral width." I understand the point that the authors are trying to make here, in defining a population. However, SNRs and spectral widths are very different things and so their values cannot be compared simply.

*R: We are not comparing the two quantities, we are trying to see if the resulting 2D distributions follow patterns determined from the existing scattering theories, i.e., from results from Figure 1.*

- Figure 7. Why have different ranges of range been used for the y-axes in panels c

and d? It would make more sense to use the same.

*R: We have preferred to focus in panel c on the echoes above MAARSY to show the convolved effect of the vertical structure and the wide receiver beam used, i.e., KAIRA's. And in panel D on the "horizontal structure" of the mean horizontal wind at PMSE altitudes. Note that the complete KAIRA SNR is already shown in Figure 5.*

- page 9 line 5. "Having defined an empirical RCS difference between the KAIRA bistatic and MAARSY monostatic of ~30 dB . . ." It would be more consistent to refer to this as an SNR difference (as in Figure 6) rather than an RCS difference. I realise that one implies the other.

*R: Done.*

- page 9, line 11. "In the case of the horizontal velocity, the estimates are consistent when a single drifting structure occurs . . ." What exactly do the authors mean by this? That the observed velocity pattern is consistent with a structure moving at a single speed

*R: Yes, we show that the localized PMSE structures, in this case, drift at the same velocity of the background horizontal wind, by comparing the parabolic behavior of SNR in range vs time, with the expected behavior given the horizontal wind obtained from the Doppler shifts.*

- page 10, line 9. "Our results also show that these PMSE structures with high Sc have a limited volume of approximately 5 - 15 km of horizontal extent in the KAIRA-MAARSY direction." Could the authors explain in more detail how they infer this - and the "cloud" separations.

*R: We are obtaining this information form Figure 7, by comparing the horizontal sections of 100 km with the MAARSY SNR behavior as a function of time. The horizontal sections have been inferred from the mean horizontal winds over the times used.*

- page 10, line 20. "KAIRA is only able to observe the red clouds . . .". There is nothing in Figure 2 that I would describe as "red". If I understand the authors correctly, I would describe these structures are "light brown", "buff", or "beige".

*R: We are now referring to them a "light brown" clouds.*

[revised manuscript text omitted]